

# Alternation of heterotrophic bacterial and archaeal production along nitrogen and salinity gradients in coastal wetlands

Gema L. Batanero[1], Andy J. Green[2], Juan A. Amat[2], Marion Vittecoq[3,4], Curtis A. Suttle[5], Isabel Reche[1,6*]

[1]Departamento de Ecología e Instituto del Agua, Universidad de Granada, 18071 Granada, Spain
[2]Departamento de Ecología de Humedales, Estación Biológica de Doñana, EBD-CSIC, 41092 Sevilla, Spain.
[3]Tour du Valat, Institut de Recherche pour la Conservation des Zones Humides Méditerranéennes, 10 Arles, 13200 France.
[4]UMR MIVEGEC, IRD, CNRS, Université de Montpellier, Montpellier, France
[5]Departments of Earth, Ocean & Atmospheric Sciences, Microbiology & Immunology, and Botany, and the Institute for the Oceans and Fisheries, University of British Columbia, Vancouver, BC, V6T 1Z4, Canada.
[6]Research Unit Modeling Nature (MNat), Universidad de Granada, 18071 Granada, Spain

*Correspondence to*: Isabel Reche (ireche@ugr.es)

**Abstract.** Coastal wetlands are valuable ecosystems with high biological productivity and diversity, which provide ecosystem services such as a reduction in the inputs of nitrogen into coastal waters, and storage of organic carbon, thus, acting as net carbon sinks. The rise of sea level as a consequence of
climatic warming will salinize many coastal wetlands, but there is considerable uncertainty about how salinization will affect microbial communities and biogeochemical processes. We analyzed prokaryotic abundance and heterotrophic bacterial and archaeal production in 112 ponds within nine coastal wetlands from the western Mediterranean coast. We determined the main drivers of prokaryotic abundance and production in these wetlands using generalized linear models (GLMs). The best GLM,
including all the coastal wetlands, indicated that the concentration of total dissolved nitrogen (TDN) positively affected the abundance of heterotrophic prokaryotes and heterotrophic archaeal production. In contrast, heterotrophic bacterial production was negatively related to TDN. This negative relationship appeared to be mediated by salinity and virus abundance. Heterotrophic bacterial production declined as salinity, and virus abundance, increased. We observed a switch from heterotrophic bacterial production
towards heterotrophic archaeal production as salinity and virus abundance increased. Our results imply



that microbial activity will change from bacterial-dominated processes to archaeal-dominated processes along with increases of nitrogen inputs and salinity. However, more studies are required to link the mineralization rates of dissolved nitrogen and organic carbon with specific archaeal taxa, to enable more accurate predictions on future scenarios of wetlands salinization and anthropogenic nitrogen
inputs.

## 1 Introduction

Wetlands have high productivity, as well as high functional and species diversity (Mitsch & Gosselink, 2000; Mitsch *et al.* 2015). Among the ecosystem services they provide, wetlands reduce nutrient loading from freshwater inflow (Verhoeven *et al.* 2006), and have been described as "the kidneys of the
landscape" (Mitsch *et al.* 2015). Coastal wetlands, in particular, are considered global reservoirs of organic carbon because of their elevated primary productivity and high sedimentation rates that, in the long-term, minimize carbon dioxide emissions to the atmosphere, acting as carbon sinks (Roehm 2005; Chmura *et al.* 2003; Geertz-Hansen *et al.* 2011). Therefore, microbial activities such as denitrification and organic carbon production and mineralization mediate these wetland functions and services.

45          The production of nitrogen fertilizer has enabled increased food production but has drastically disrupted the nitrogen cycle by doubled input of nitrogen to the Earth´s surface, and led to environmental problems such as eutrophication (Gruber & Galloway, 2008; Canfield *et al.* 2010). This change likely exceeds all other human effects on nutrient cycles (Gruber & Galloway 2008; Schlesinger 2009) but has received relatively less attention compared to the carbon cycle (Battye *et al.* 2017). A
substantial fraction of this nitrogen, mostly derived from agricultural lands, is transported to rivers and groundwater through runoff and is lost through emissions of ammonia and other nitrogen compounds by denitrification (Schlesinger 2009; Battye *et al.* 2017). In many estuaries, natural or constructed wetlands modify the nitrogen loading into coastal areas and reduce coastal eutrophication (Walton et al. 2015). These transitional waters include saline wetlands such as natural marshes and multi-pond solar salterns
(Razinkovas-Baziukas and Povilanskas, 2012).

         Multi-pond solar salterns are present along the coasts of the Mediterranean and Portugal since the Phoenicians, who used evaporation procedures dating from the Emperor Huang era about 2,500





years B.C. to obtain salt (Baas-Becking 1931; García-Vargas & Martínez-Maganto 2006). Coastal
saline wetlands also have great importance for waterbird conservation and sustainable aquaculture

(Britton & Johnson 1987; Sánchez et al. 2006; Athearn *et al.* 2009; Walton *et al.* 2015). Most studies of
microbial ecology in coastal wetlands have focused on extremophile microorganisms and their
bioenergetic and diversity constraints (Antón *et al.* 1999; Pedrós-Alió *et al.* 2000; Oren 2001;
Casamayor *et al.* 2002; Gasol *et al.* 2004; Ghai *et al.* 2011); much less is known about moderately
halophilic bacteria and archaea in oligo- to eu-saline waters (Hahn, 2006; Herbert *et al.* 2015).

65       The discovery that archaea are not exclusively extremophiles (DeLong 1992; Fuhrman *et
al.* 1992; Karner *et al.* 2001) led to studies showing that archaea are abundant, ubiquitous and diverse,
with essential roles in carbon and nitrogen cycling (Herndl *et al.* 2005; Justice *et al.* 2012; Offre *et
al.* 2013), including mineralizing organic matter under oxic conditions in the ocean (Herndl *et al.* 2005;
Ingalls *et al.* 2006). The organo-heterotrophic nature of some archaeal groups has been shown for

specific carbohydrates (Lazar *et al.* 2016) and dissolved proteins (Orsi *et al.* 2016). Archaea also
oxidize ammonia, and perform denitrification under anoxic or suboxic conditions (Francis *et al.* 2007;
Offre *et al.* 2013) with potential implications for nitrogen removal in wetlands (You *et al.* 2009).
However, the prevalence of heterotrophic metabolism in archaea and its biogeochemical relevance are
poorly studied. As well, archaeal heterotrophic production patterns along gradients of dissolved organic

carbon and nitrogen, and salinity, remain completely unexplored in the literature.

Currently, most wetlands are salinizing as a consequence of climatic warming and freshwater
extraction (Herbert *et al.* 2015; Jeppesen et al. 2015). Coastal wetlands are particularly affected since
the rise of the sea level is introducing seawater high up into estuaries (Herbert *et al.* 2015; Schuerch *et
al.* 2018). Salinization alters water and sediment chemistry and changes biogeochemical reactions

(Herbert *et al.* 2015). Although there is still considerable uncertainty about how salinization affects
wetland biogeochemistry, it promotes the release of inorganic nitrogen ($NH_4^+$) and phosphorus ($PO_4^{3-}$),
with implications for internal wetland eutrophication (Ardón *et al.* 2013; Weston *et al.* 2006). Besides,
salinization increases the generation of toxic sulfides (Lamers *et al.* 1998), which play an essential role
in wetland N cycling, inhibiting the final steps of denitrification, and resulting in the emission of

nitrogen oxides (Brunet and García-Gil 1996, Laverman *et al.* 2007). Overall, salinization disrupts





many ecosystem services provided by wetlands, such as their capacity to store organic carbon (Weston *et al.* 2011; Luo *et al.* 2017) and remove nitrogen from water (Herbert *et al.* 2015; Franklin *et al.* 2017).

90       The projected scenario of climatic warming includes both wetland salinization and more nitrogen inputs. These environmental risks interact, with increased nitrogen boosting organic carbon mineralization and reducing carbon storage in coastal marshes (Deegan *et al.* 2012), while salinization can affect microbial nitrogen removal (Franklin et al. 2017). Our work describes patterns in microbial abundance and heterotrophic bacterial and archaeal production in a broad set of semi-natural and constructed wetlands covering gradients of salinity, dissolved organic carbon, and nitrogen along the

Western Mediterranean. Furthermore, we identified the main drivers of these patterns and discussed potential changes in prokaryotic heterotrophic production under future scenarios.

## 2 Materials and methods

### 2.1 Study sites

We sampled 112 saline ponds from nine coastal wetlands during late spring-midsummer of 2011, 2012,

and 2013 (Supplementary Table S1). The coastal wetlands were the Odiel marshes (OdielM) at the mouth of the Odiel river, the Veta la Palma (VPalma) fishponds at the mouth of the Guadalquivir river, the Cabo de Gata (CGata) salt pans, Santa Pola (SPola) and El Hondo (Hondo) at the mouth of the Vinalopó river, and the Ebro Delta (EbroD) ponds at the mouth of the Ebro river in Spain, Salin-de-Giraud and Saintes-Maries-de-la-Mer at the mouth of the Rhône (Camargue, France), the Molentargius,

Santa Guilla, and Santa Caterina ponds (Sardinia, Italy), and Thyna solar salterns (Sfax, Tunisia) (Fig.1a). All the study ponds are located in semiarid or arid areas, under typical Mediterranean climatic conditions, covering large gradients of salinity from hypersaline (solar salt pans) to oligo-mesohaline (brackish) waters (Fig.1b), and microbial productivity (Fig.1c). Solar salterns are often multi-pond systems connected for commercial salt production, which creates a strong salinity gradient from

evaporation ponds to crystallizer ponds (Fig.1b). They are natural laboratories across which microbial communities can be compared (Fig.1c). The brackish wetlands that we studied are also crucial for



sustainable aquaculture and waterfowl conservation (Britton & Johnson 1987; Walton *et al.* 2015). Complete location, sampling dates, and physical-chemical and microbiological details for each pond are in Supplementary Table S1.

**2.2 Chemical analyses**

We recorded salinity with a multi-parameter probe (HANNA HI 9828). Water samples with salinity higher than 70 ppt were diluted with Milli-Q water until they were in the operating range of the probe. We measured total nutrient concentrations in unfiltered samples, while we filtered the samples through pre-combusted 0.7-µm pore-size Whatman GF/F glass-fiber filters for dissolved nutrient analysis. We

measured total phosphorus (TP), and total dissolved phosphorus (TDP) using the molybdenum blue method (Murphy & Riley, 1962) after persulfate digestion (30 min, 120ºC). We analyzed total nitrogen (TN) and total dissolved nitrogen (TDN) by high–temperature catalytic oxidation (Álvarez-Salgado and Miller 1998) using a total nitrogen analyzer (Shimadzu TNM-1), and potassium-nitrate standards. We analyzed dissolved organic carbon (DOC) concentration as non-purgeable organic carbon by high–

temperature catalytic oxidation using a total organic carbon analyzer (Shimadzu TOC-V CSH). DOC samples were pre-filtered through pre-combusted Whatman GF/F filters (2 h at 500ºC) and acidified with $H_3PO_4$ (final pH<2). We calibrated the instrument using a four-point standard curve of potassium hydrogen phthalate. We purged the samples with phosphoric acid for 20 min, and we set up three to five injections for each sample.

**2.3 Biological analyses**

We determined chlorophyll-*a* (Chl-*a*) concentration by extracting Whatman GF/F filters in 95% methanol in the dark for 24 h at 4 ºC (APHA 1992) through which 50 to 2000 ml of water was filtered. Absorbance was measured at 665 nm and 750 nm using a Perkin Elmer Lambda 40 spectrophotometer.

Samples for determining the abundances of prokaryotes (PA) and cyanobacteria (CyA) by flow
cytometry (Gasol and del Giorgio, 2000) were collected in cryovials, fixed with 1% paraformaldehyde and 0.05% glutaraldehyde in the dark (30 min at 4 ºC), then frozen in liquid nitrogen and stored at -80 ºC until analysis. Once defrosted, the samples were diluted (≥10-fold) with Milli-Q water to avoid the



electronic coincidence of the prokaryotic cells. We counted the PA and CyA in a FACScalibur flow cytometer with a laser emitting at 448nm and a suspension of yellow-green 1-μm latex beads

(Polysciences) per sample as an internal standard. The PA samples were stained in the dark (10 min) with a 10 μM DMSO solution of Sybr Green I stain (Molecular Probes), run at low speed for 2 min, and detected in bivariate plots of side scatter (SSC) vs. FL1 (Green fluorescence). The CyA samples were run at high speed for 4 min and detected in bivariate plots of side scatter (SSC) vs. FL3 (Red fluorescence). We obtained the heterotrophic prokaryotic abundance (HPA) by subtracting the

cyanobacteria abundance (CyA) from prokaryotic abundance (PA).

Samples for virus abundance (VA) were collected in cryovials, fixed with glutaraldehyde at a final concentration of 0.5 % (15 min at 4ºC) in the dark, frozen in liquid nitrogen, and stored at -80ºC until analysis by flow cytometry (Brussard *et al.*, 2010). Before analysis, we diluted the samples ≥ 100-fold with TE-buffer pH 8.0 (10 mM Trishydroxymethyl-aminomethane, 1 mM

ethylenediaminetetraacetic acid) to avoid the electronic coincidence in virus particle counts. We stained the VA samples with a working solution (1:200) of SYBR Green I (10,000X concentrate in DMSO, Molecular Probes) for 10 min in the dark and then kept them at -80º until counting. We added fluorescent microspheres (FluoSpheres carboxylate modified yellow-green fluorescent microspheres; 1.0 μm diameter) as an internal standard. We acquired the data in log mode and detected their signature

in bivariate plots of side scatter (SSC) vs. FL1 (Green fluorescence). We processed the flow cytometry data using BD CellQuest Pro software.

Heterotrophic bacterial production (BP) and archaeal production (ArP) were measured by $^3$H-Leucine incorporation into proteins (Smith and Azam, 1992). The heterotrophic archaeal activity was measured using erythromycin to inhibit bacterial activity, as proposed by Yokokawa *et al.* (2012).

Although the specificity of erythromycin to selectively inhibit bacteria has recently been questioned for the open ocean (Frank *et al.* 2016), it appears to have better efficiencies (ca. 80%) in water of higher salinity (Oren *et al.* 1990; Pedrós-Alió *et al.* 2000) and for specific functional groups as nitrifiers, particularly Firmicutes (Du *et al.* 2016). For each pond, two sets of three replicate (1.5 ml) and two trichloroacetic-acid (TCA, 50%)-killed (final concentration10%) samples were incubated at *in*

*situ* temperature with 54.6 nM or 58.4 nM leucine (1:2 hot: cold v/v) for 2 to 5 h. One of each set of 5





samples also contained 10 µg ml$^{-1}$ of erythromycin (final concentration) to inhibit bacterial production, and obtain heterotrophic production attributed mostly to archaea. We added TCA at a final concentration of 10% to end the incubations. In the laboratory, the samples were centrifuged (15366 g for 10 min), rinsed with TCA (5%), vortexed, and centrifuged again. Finally, we added 1.5 ml of liquid

scintillation cocktail (Ecoscint A) to each sample and determined the incorporated radioactivity using an auto-calibrated scintillation counter (Beckman LS 6000 TA).

**2.4 Statistical analyses**

To determine the main drivers of the observed microbial patterns, we carried out four sets of generalized linear models (GLMs). In the first set of GLMs, the dependent microbial variables were

heterotrophic prokaryotic abundance (HPA), cyanobacterial abundance (CyA), heterotrophic bacterial production (BP), and heterotrophic archaeal production (ArP). The predictor variables selected were site (i.e., each wetland complex as a categorical variable), and as continuous variables, salinity, total dissolved nitrogen (TDN), and total dissolved phosphorus (TDP). When necessary, log and square-root transformations were applied to the dependent and predictor variables to improve the fit to a normal

distribution, and to avoid heteroscedasticity. In the second set of GLMs, we also included the concentration of DOC as a predictor variable but did not include the Camargue site since this variable was not measured there. In the third set of GLMs, we determined the best predictors of virus abundance (dependent variable) in the six sites (Odiel M, VPalma, CGata, SPola, Hondo, and EbroD) where this variable was available. Nutrients were not included as predictors because viruses are mostly dependent

on host (bacterial or archaeal) density. We considered the study site as a categorical variable, and salinity, PA, and CyA as continuous variables. To determine the potential effect of viruses on the other microbial components, we performed the fourth set of GLMs in which VA is considered as a predictor variable and BP and ArP as dependent variables. We selected the model with the smallest value of the Akaike Information Criterion (AIC) as the best model for each dependent variable considered. More

details of alternative models with ΔAIC < 2.0 are in the Supplementary Tables S2-S6. All analyses were performed using Statistica 7.0 software.



## 3 Results

The study ponds represented a wide range of environmental conditions. Salinity ranged more than four orders of magnitude, from 0.22 to 343 ppt (Table 1), spanning oligo- to hyper-saline conditions; the highest median value was in EbroD ponds, and the lowest median value in El Hondo ponds (Fig. 2a). DOC concentrations ranged from 0.24 to 5.76 mmol-C $l^{-1}$ (Table1), with the highest median concentration in EbroD ponds (as for salinity) and the lowest in CGata ponds (Fig. 2b). TDN concentrations ranged from 0.02 to 0.62 mmol-N $l^{-1}$ (Table 1), with EbroD ponds also showing the highest median TDN concentration and SPola salt pans the lowest values (Fig. 2c). TDP concentrations ranged from 0.50 to 40.09 μmol-P $l^{-1}$ (Table 1), with the highest median value at OdielM wetlands and the lowest median at CGata salt pans (Fig. 2d).

Biological variables also ranged widely. Prokaryotic heterotrophic abundances (PHA) ranged almost three orders of magnitude, from 0.35 x$10^6$ to 252 x $10^6$ cells ml$^{-1}$ (Supplementary Table S1), with the highest median value in Tyna salt pans and the lowest in CGata salt pans (Fig. 3a). The range in cyanobacterial abundances was even higher, ranging from 0.01 to 38105 (x$10^3$) cells ml$^{-1}$ (Supplementary Table S1; Fig. 3b). Heterotrophic bacterial production ranged from 19 to 3007 pmoles of leucine $l^{-1}h^{-1}$ (Supplementary Table S1; Fig. 3c) with the highest median value at SPola salt pans and the lowest at Thyna salt pans. Heterotrophic archaeal production ranged from 56 to 2290 pmoles of leucine $l^{-1}h^{-1}$ with the highest median value at EbroD ponds and the lowest at Camargue ponds. (Supplementary Table S1; Fig. 3d). Chlorophyll-a concentrations ranged more than 10000-fold from 0.04 to 617.41 μg $l^{-1}$ (Table1), with the highest median value at Thyna salt pans and the lowest at EbroD ponds (Fig. 3e).

In the first set of GLMs, the best model for heterotrophic prokaryotic abundance included the TDN concentration and the wetland site (categorical) as predictors (Table 2). The residuals of heterotrophic prokaryotic abundance (once the site was controlled for) were significantly and positively related to TDN (Fig. 4 a). Alternative (less explanatory) models also included salinity or TDP (Table supplementary S2). The best GLM for cyanobacterial abundance included the site as a categorical variable and TDN and TDP as continuous variables, with a significant positive partial relationship with TDN and negative partial relationship with TDP (Table 2). In the case of heterotrophic bacterial





production, the best GLM included TDN and the wetland site as predictors (Table 2). The residuals of heterotrophic bacterial production (once the site was controlled for) were significantly and negatively related to TDN (Fig. 4 b). Indeed, the higher the TDN concentration, the lower the bacterial production, irrespectively of the site. Alternative (less explanatory) models also included salinity or TDP (Supplementary Table S2). For archaeal heterotrophic production, the best GLM also included TDN and

site as predictors (Table 2). In contrast to heterotrophic bacterial production, the residuals of heterotrophic archaeal production (once the site was controlled for) were significantly and positively related to TDN concentration (Fig. 4 c). Indeed, the higher the TDN, the higher the archaeal production, irrespective of site. Alternative (less explanatory) models also included salinity or TDP (Supplementary Table S2).

In the second set of GLMs, we also included the concentration of DOC as a predictor variable but excluded the Camargue site since these data were not available. This predictor variable only affected the results previously exposed to the heterotrophic prokaryotic abundance and the heterotrophic bacterial production (Table supplementary S3). In this second analysis, heterotrophic prokaryotic abundance was affected negatively by salinity and positively by DOC concentration, while depending

on the site, DOC concentration negatively affected bacterial production.

In the third set of GLMs, we determined the best GLM for the virus abundance, excluding nutrients as predictors (Table 3). This model included salinity as a continuous predictor and site as a categorical predictor. We observed a positive relationship between the residuals of virus abundance (once the site was controlled for) and salinity (Figure 5). Alternative (less explanatory) models also

included heterotrophic prokaryotic and cyanobacterial abundances (Table supplementary S4).

Finally, in the fourth set of GLMs, we included viral abundance as a predictor of heterotrophic bacterial and archaeal production (Table supplementary S5), but only considered the following sites: Odiel M, VPalma, CGata, SPola, Hondo, and EbroD. The best GLM for heterotrophic bacterial production included salinity and virus abundance as predictors (Table 4). Once the site effect was

controlled for, we observed a significant negative relationship between heterotrophic bacterial production and salinity (Figure 6a), as well as viral abundance (Figure 6b). On the other hand, the best GLM for heterotrophic archaeal production was coherent with the first set of GLMs, which included all



the saline wetlands; the best GLM included TDN and site as predictors (Table supplementary S5). This result confirms the robust relationship between TDN and heterotrophic archaeal production in the study

wetlands.

## 4 Discussion

The best generalized linear model (GLM), including all the coastal wetlands, indicated that the concentration of TDN positively affected the abundance of heterotrophic prokaryotes and heterotrophic archaeal production. TDN was consistently associated with greater archaeal heterotrophic production,

even considering the alternative GLMs with other predictor variables, such as DOC concentration and viral abundance (VA). In contrast, TDN affected negatively heterotrophic bacterial production, but this result changed when DOC and VA were also considered as predictor variables in alternative GLMs. The negative relationship between TDN and heterotrophic bacterial production appeared to be mediated by salinity and VA. Heterotrophic bacterial production declined as salinity and viruses increased;

therefore, there was a switch from heterotrophic bacterial production to heterotrophic archaeal production as salinity and VA increased. Archaea appeared to be the main microorganisms processing TDN in the most saline wetlands.

In this across-wetlands study, salinity did not directly affect heterotrophic archaeal production, whereas total dissolved nitrogen was consistently the best predictor, irrespectively of the variables

considered, in the generalized linear models. Historically, archaea were associated with extreme environments, such as salterns, where salt concentrations are close to saturation (Oren, 1994, 2011; Antón et al. 1999). Now, we know that archaea occur in a wide range of environments and perform diverse functions in N cycling, including nitrification and denitrification (Francis et al. 2007, Offre et al. 2013). In our study, ammonia oxidation by archaea during nitrification likely is not a significant process

due to the high concentrations of dissolved nitrogen in most wetlands, particularly in the Ebro Delta (Fig. 2). Ammonia-oxidizing archaea (AOA) are abundant in marine systems (Könneke et al. 2005, Wuchter et al. 2006, Lam et al. 2007), some lakes (Jiang et al. 2009), and wetlands (Sims et al. 2010). The relative contribution of AOA to ammonium oxidation is still uncertain, but they appear to dominate over ammonia-oxidizing bacteria (AOB) at low ammonium concentrations (Martens-Habbena et al.





2009). Sims *et al.* (2010) also observed that AOA were dominant over ammonia-oxidizing bacteria (AOB) in freshwater oligotrophic wetlands.

On the other hand, our results emphasize that heterotrophic archaeal activity, measured as the uptake of 3H-leucine, was coupled to total dissolved nitrogen concentration (Fig. 4c). Denitrification is primarily a heterotrophic, microaerophilic, or anoxic process widespread along salinity gradients, which

can be done by archaea (Zumft, 1997). Coastal wetlands provide optimal conditions for archaeal denitrification due to their low oxygen concentrations, as a result of high salinities, and high concentrations of organic matter and nitrate (Seitzinger, 1988; Seitzinger et al. 2006). Furthermore, Orsi et al. (2016) demonstrated the organo-heterotrophic nature of some archaeal groups, which take up dissolved proteins. Therefore, heterotrophic archaeal metabolism via denitrification and protein

assimilation affects the nitrogen cycle ameliorating eutrophication of recipient coastal systems. However, a more detailed analysis of functional genes of N processing, beyond heterotrophic production, is needed to quantitatively assess the contribution of archaea to nitrogen cycling in coastal wetlands.

Early works emphasized the importance of organic carbon derived from phytoplankton as a

critical driver of heterotrophic prokaryotic (bacterial + archaeal) production in diverse aquatic ecosystems (e.g., Cole *et al.* 1988; Baines and Pace 1991; Ortega-Retuerta *et al.* 2008). However, in this study, we did not find difference between the best GLMs for heterotrophic archaeal production with and without the concentration of dissolved organic carbon (DOC) (Table S3). Indeed, the total pool of DOC concentration was not a predictive variable at least for heterotrophic archaeal production. In saline

wetlands, extreme evapoconcentration (Batanero *et al.* 2017) and high loads of organic matter from freshwater inflows (Walton *et al.* 2015) result in high concentrations of DOC in oligo- to eusaline wetlands, likely ensuring plenty DOC for heterotrophic metabolism. Similarly, TDP was not included as a predictor variable in the best GLMs for the different microbial variables, except for the abundance of cyanobacteria (CyA). Bacterioplankton growth is assumed to be limited by the availability of

phosphorus in many aquatic ecosystems (Cotner *et al.* 1997; Smith and Prairie 2004; Ortega-Retuerta *et al.* 2007). However, more recently, Cotner *et al.* (2010) demonstrated that bacteria exhibit high flexibility in their P content and stoichiometry, questioning the ranges of P-limitation. Our data suggest



that neither phosphorus nor dissolved organic carbon appear to affect significantly the prokaryotic activity in the full range of saline wetlands studied.

In the study wetlands, heterotrophic bacterial production (BP) appeared to be controlled by viral abundance (VA) and salinity (Fig. 6), providing an indirect explanation for the negative correlation between total dissolved nitrogen and BP (Fig. 4b). The results showed that, within a given wetland complex, VA increased significantly with salinity (Table 3, Fig. 5), and negatively affected the BP (Table 4, Fig. 6b), but did not have a significant effect on heterotrophic archaeal production (Table

supplementary S5). These findings suggest that most of the viruses were likely bacteriophages that caused significant bacterial mortality in these extremely saline wetlands. Thus, external factors such as evaporation would trigger an increase in VA, which in turn would increase bacterial mortality, while archaeal communities would be more resilient to such changes. Guixa-Boixereu *et al.* (1996) observed the highest prokaryotic losses by viral lysis at the highest salinities in solar salterns. Another

complementary explanation is that as the salinity increases, there is a coupled decline in oxygen solubility, promoting suboxic conditions. High salinity and low oxygen may be more energetically favourable conditions for some halophilic archaeal groups (Oren, 2001).

The projected scenario of climatic warming includes wetland salinization due to sea-level rise and decreases of freshwater discharges from rivers, which will allow marine waters further into the

estuaries, salinizing the surrounding wetlands (Herbert *et al.* 2015; Schuerch *et al.* 2018). Human population growth and crop production are leading to more nitrogen inputs (Mulholland *et al.* 2008; Mekonnen & Hoekstra, 2015). These two environmental risks can interact, affecting the potential services of these coastal wetlands. Eutrophication can boost the mineralization of organic carbon, reducing the carbon storage capacity of coastal marshes (Deegan *et al.* 2012), and salinization can affect

the bacterial capacity to remove nitrogen (Franklin et al. 2017). Our results imply that microbial activity will change from bacterial dominated processes to archaeal-dominated processes along with the increases of nitrogen and salinity. More studies linking the processing rates of dissolved nitrogen and organic carbon with specific archaeal taxa will allow more accurate predictions in the next scenarios of global change.




## 4 Conclusions

The concentration of total dissolved nitrogen appeared to determine the abundance of heterotrophic prokaryotes and, mainly, the heterotrophic archaeal production in the suite of coastal wetlands studied. In contrast, total dissolved nitrogen affected negatively to heterotrophic bacterial production. We think

that this negative relationship between total dissolved nitrogen and heterotrophic bacterial production appeared to be mediated by salinity and virus abundance. Heterotrophic bacterial production declined as salinity and viruses increased; therefore, there was a switch from heterotrophic bacterial production in wetlands with lower nitrogen content and salinity to heterotrophic archaeal production in wetlands with higher nitrogen and salinity. Archaea appeared to be the main prokaryotes processing nitrogen in the

most saline wetlands. Therefore, changes in nitrogen inputs and salinities associated with global change can alter prokaryotic functional groups and, consequently, the ecosystem services provided by coastal wetlands.

## Data availability

Additional figures and tables can be found in the supplementary information.


## Author contributions

I.R., A.J.G., J.A.A., M.V. and C.A.S. designed this study; G.L.B., A.J.G., J.A.A., M.V., and I.R. performed the samplings and most of biological and chemical analysis. G.L.B. and C.A.S. analyzed the virus abundance. G.L.B., A.J.G., and I.R. analyzed the data and wrote the manuscript with input from

all the authors.

## Competing interests

The authors declare that the research was conducted in the absence of any commercial or financial relationships that could be construed as a potential conflict of interest.




**Acknowledgements**

This research was funded by the projects the projects FLAMENCO (CGL2010-15812) and HERA (CGL2014-52362-R) of the Spanish Ministry of Economy and Competitiveness, the Modeling Nature
Scientific Unit (UCE.PP2017.03), Consejería de Economía, Conocimiento, Empresas y Universidad and European Regional Development Fund (ERDF), ref. SOMM17/6109/UGR, and a PhD fellowship FPI (Formación del Personal Investigador: BES-2011-043658) from the Spanish Government to GLB. We specially thank Alejandra Fernández and Eulogio Corral for their help in the field and laboratory. We also thank the facilities and technical staff of Odiel Marshes, Veta la Palma (Doñana National Park), El
Hondo, Santa Pola, Ebro Delta, Tour du Valat (Camargue), Molentargius (Sardinia), and Université de Gabês (Tunisia). We also thank the logistic support of Josefa Antón at the University of Alicante, Arnaud Béchet at Tour du Valat, and M.A. Chokri and S. Selmi in the Tunisian site.

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





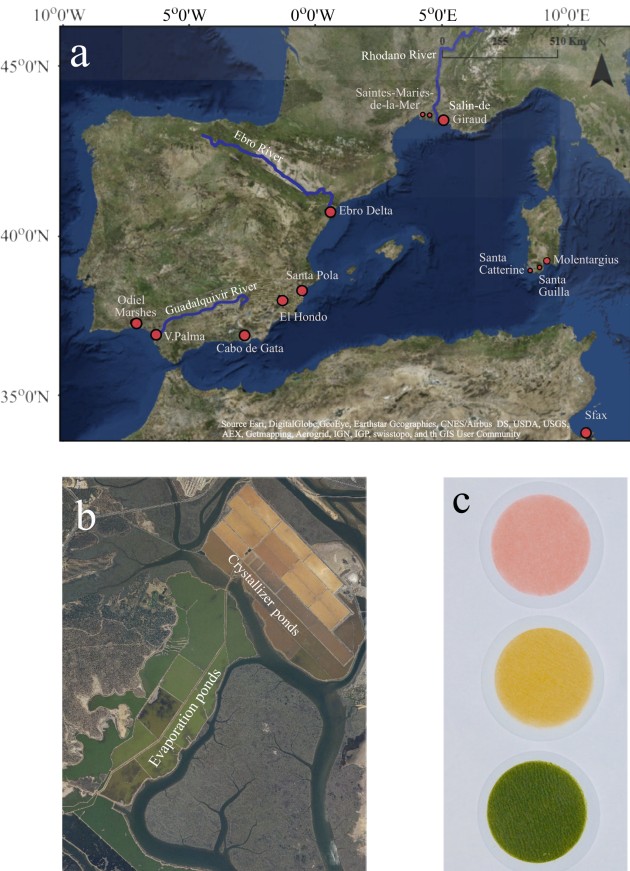

**Figure 1**. Map obtained using ArcGIS including the locations of the coastal wetlands studied (a): Odiel marshes (OdielM), Veta la Palma, (VPalma), Cabo de Gata (CGata), Santa Pola (SPola), El Hondo (Hondo), Ebro Delta (EbroD), Salin-de-Giraud and Saintes-Maries-de-la-Mer (Camargue), Molentargius, Santa Guilla and Santa Catterine (Sardinia), and Sfax (Sfax). Imagery map credit: Esri, DigitalGlobe, Earthstar Geographics, CNES/Airbus DS, GeoEye, USDA FSA, USGS, Aerogrid, IGN, IGP, and the GIS User Community. (b) Orthophoto of the Odiel marshes including evaporation and

crystallization ponds; Orthophoto credit: Junta de Andalucía. Consejería de Agricultura, Ganadería, Pesca y Desarrollo Sostenible (ámbito de Desarrollo Sostenible). http://www.juntadeandalucia.es/medioambiente/site/rediam/
(c) G/F glass-fiber filters illustrating the pigment diversity of these ponds.





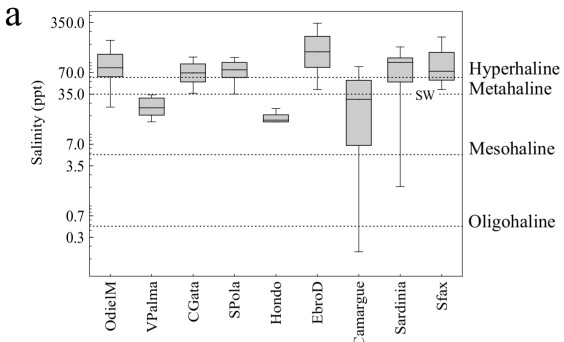

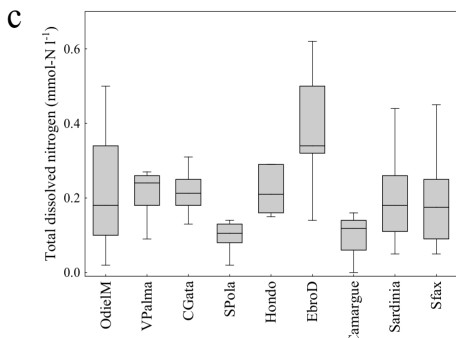

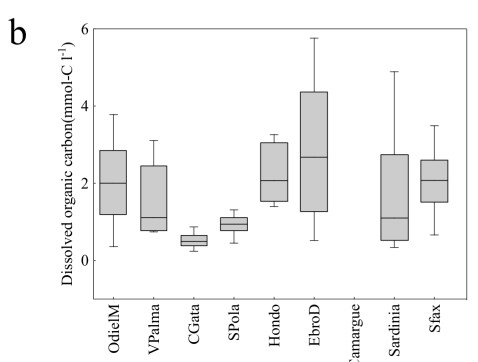

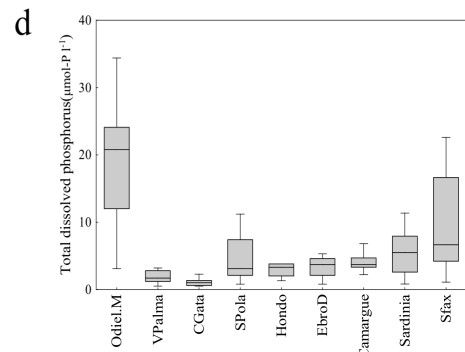


**Figure 2.** Summary of raw data for physico-chemical parameters in the western Mediterranean basin, showing salinity (a), total dissolved organic carbon (b), total dissolved nitrogen (c) and total dissolved phosphorus (d) for each saline wetland studied. Lines = Median values. Boxes = 25% and 75% percentiles and whiskers = non-outlier range. Outliers are shown as dots and extreme points as asterisks.


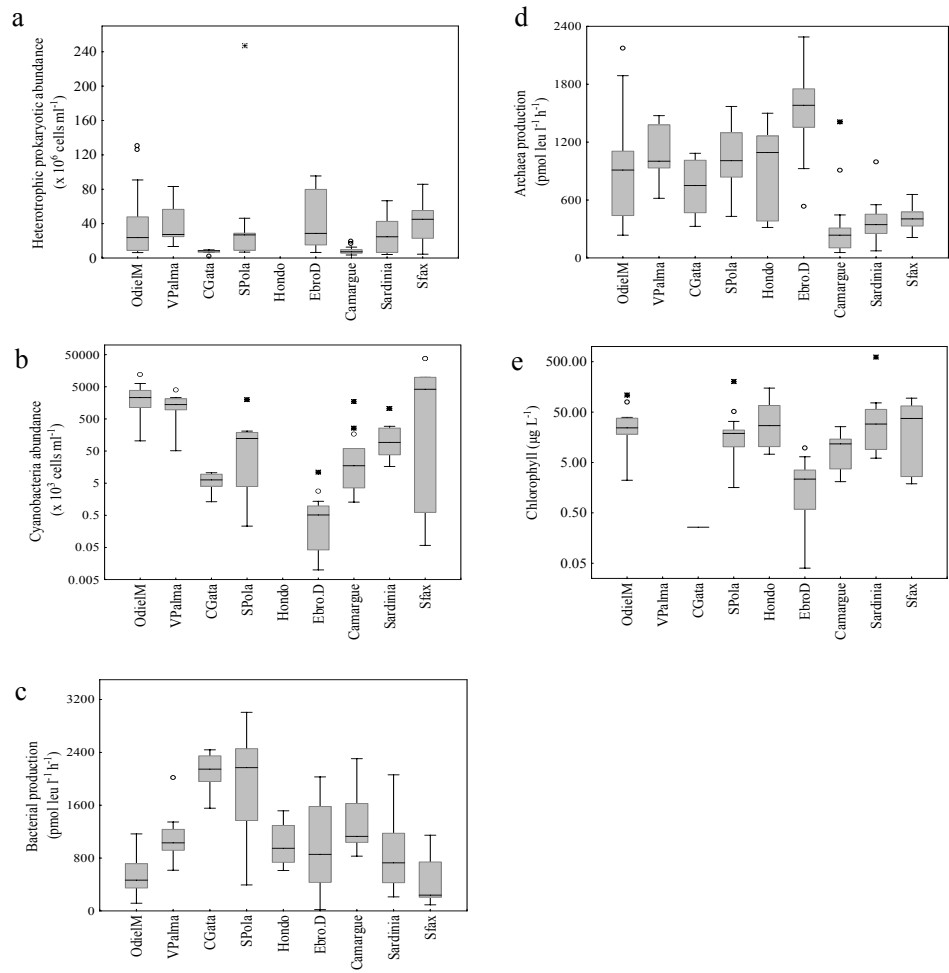


**Figure 3.** Summary of raw abundance and production of heterotrophic prokaryotes and cyanobacteria in the western Mediterranean basin, showing heterotrophic prokaryotic abundance (a), cyanobacteria abundance (b), bacterial production (c), archaea production (d) and chlorophyll concentration (e) for each saline wetland studied. Lines = Median values. Boxes = 25% and 75% percentiles and whiskers= non-outlier range. Outliers are shown as dots and extreme points as asterisks.







a
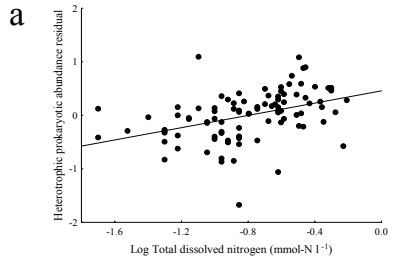

b
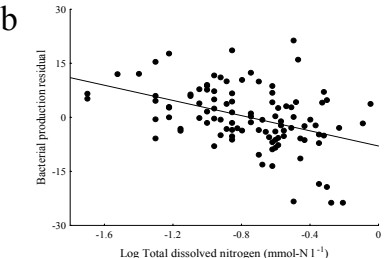

c
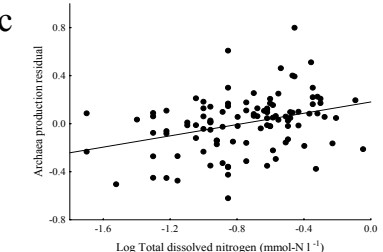

**Figure 4.** Linear regressions between heterotrophic abundance and prokaryotic production and total dissolved nitrogen (TDN). Partial effects are presented for relationships between (a) heterotrophic prokaryotic abundance, (b) bacterial production and (c) archaea production and TDN, based on the best models shown in Table 2. In each case the Y variable represents the residuals taken from the selected model after fitting the other predictor variables.




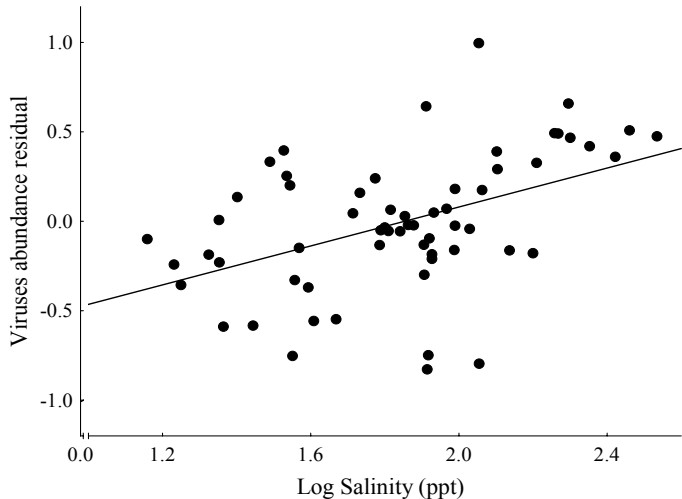

**Figure 5.** Partial effect for relationship between virus abundance and salinity based on the best models shown in Table 3.



a

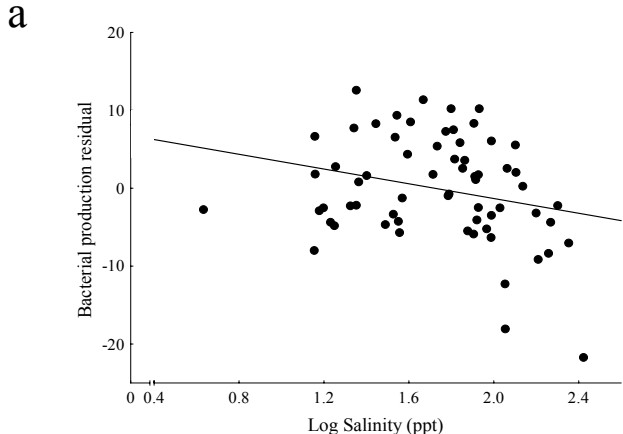

b

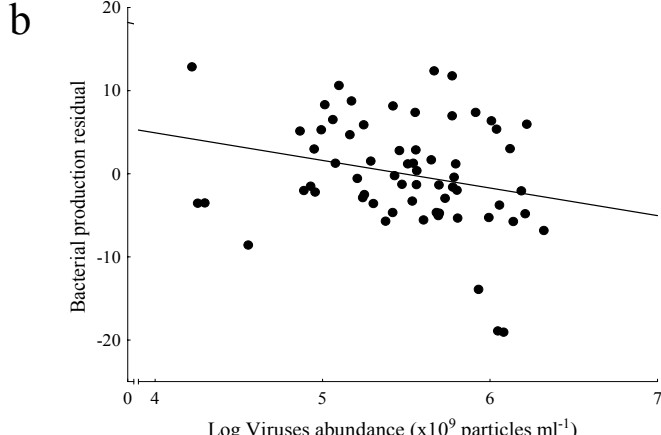

**Figure 6.** Linear regressions between bacterial production and their biological and physico-chemical predictors. Partial effects are shown for relationships between (a) salinity (b) virus abundance, based on the best models shown in Table 4. In each case the Y variable represents the residuals taken from the selected model after fitting the other predictor variables.



Table 1. Ranges of the basic physicochemical variables, total dissolved nitrogen (TDN), total dissolved phosphorus (TDP), dissolved organic carbon (DOC), concentration of chlorophyll-*a* (Chl-*a*) and virus abundance (VA) for each set of ponds in the study wetlands. The complete data set is in the Supplementary Table S1.

| Location Sites (acronyms) | Nº Ponds | Temperature (ºC) | Salinity (ppt) | TDN (mmol-N l$^{-1}$) | TDP (μmol-P l$^{-1}$) | DOC (mmol-C l$^{-1}$) | Chla (μg l$^{-1}$) | VA (x10$^9$ particles ml$^{-1}$) |
|---|---|---|---|---|---|---|---|---|
| Odiel Marshes (OdielM) Huelva, Spain | 19 | 20.1-25.4 | 23.1 -197.8 | 0.02-0.50 | 3.12- 48.06 | 0.36-3.78 | 2.22 - 108.99 | 0.17 - 3.07 |
| Veta la Palma, Doñana (VPalma) Sevilla, Spain | 10 | 27-33.5 | 14.4-34.3 | 0.09-0.27 | 0.51-12.27 | 0.74-3.11 | n.d. | 0.2 - 1.14 |
| Cabo de Gata (CGata) Almeria, Spain | 6 | 24.9-27.6 | 36.1-115.6 | 0.13-0.31 | 0.51-2.28 | 0.24-0.87 | n.d. | 0.04 – 0.12 |
| Santa Pola (SPola) Alicante, Spain | 14 | 22-27.3 | 35.1-162.1 | 0.02-0.32 | 0.81-11.22 | 0.45-2.91 | 1.59 - 201.83 | 0.02 – 1.11 |
| El Hondo (Hondo) Alicante, Spain | 7 | 24.2-30.6 | 4.3-22.0 | 0.15-0.29 | 1.31-8.96 | 1.4-3.26 | 7.3 - 149.63 | 0.35 – 1.10 |
| Ebro Delta (EbroD) Tarragona, (Spain) | 13 | 23.8-32.9 | 40.6-343 | 0.14-0.62 | 0.81-5.35 | 0.52-5.76 | 0.04- 9.7 | 0.09 – 1.69 |
| Salin-de-Giraud, Saintes-Maries-de-la-Mer (Camargue), France | 14 | 19.1-30.3 | 0.2-194.7 | 0.04-0.35 | 2.22-6.89 | n.d. | 2.08-25.69 | n.d. |
| Molentargius, Santa Guilla and Santa Caterina (Sardinia), Italy | 13 | 25.4-33.4 | 2.0-238.8 | 0.05-0.44 | 0.81-11.35 | 0.34-4.89 | 6.09 - 617.41 | n.d. |
| Sfax (Sfax) Sfax, (Tunisia) | 12 | 24.5-29.8 | 40.6-220.2 | 0.05-0.45 | 2.67-40.94 | 0.41-3.49 | 1.89- 94.58 | n.d. |






Table 2. Summary of the best-generalized linear models (GLMs) (according to AIC) selected for the prokaryotic abundance and heterotrophic production. Columns show the estimates, standard errors, Wald statistic and P values for the selected predictor variables. The complete set of alternative (less explanatory) models with δAIC< 2.0 are shown (Supplementary Table S2).

| Dependent Variables | Predictor Variables | Estimate | Standard Error | Wald Stat. | p-values |
|---|---|---|---|---|---|
| Prokaryotic heterotrophic abundance | Intercept | 2.068672 | 0.013074 | 25035.27 | 0.000000 |
| | TDN | 0.109782 | 0.015932 | 47.48 | 0.000000 |
| | Site | --- | --- | 43.76 | 0,000000 |
| Cyanobacteria abundance | Intercept | 1.799727 | 0.105138 | 293.0189 | 0.000000 |
| | Site | --- | --- | 106.1193 | 0.000000 |
| | TDP | -0.267744 | 0.080808 | 10.9782 | 0.000922 |
| | TDN | 0.181139 | 0.087963 | 4.2406 | 0.039468 |
| Heterotrophic Bacterial production | Intercept | 3.044352 | 0.075297 | 1634.684 | 0.000000 |
| | TDN | -0.492054 | 0.089367 | 30.316 | 0.000000 |
| | Site | --- | --- | 107.336 | 0.000000 |
| Heterotrophic Archaeal production | Intercept | 1.125856 | 0.021440 | 2757.573 | 0.000000 |
| | TDN | 0.139250 | 0.028357 | 24.115 | 0.000001 |
| | Site | --- | --- | 112.124 | 0.000000 |





Table 3. Summary of the best-generalized linear model (according to AIC) selected for the virus
abundance in the sites: Odiel M, VPalma, CGata, SPola, Hondo, and EbroD (see Figure 1 and Table 1).
Columns show the estimates, the standard errors, the Wald statistics, and the p-values for the predictor
variables. The complete set of alternative (less explanatory) models with δAIC< 2.0 are shown in
Supplementary Table S4.

| Dependent Variables | Predictor Variables | Estimate | Standard Error | Wald Stat. | p-values |
|---|---|---|---|---|---|
| Virus abundance | Intercept | 1.316355 | 0.053246 | 611.1956 | 0.000000 |
| | Salinity | 0.204842 | 0.028806 | 50.5662 | 0.000000 |
| | Site | --- | --- | 117.4885 | 0.000000 |






Table 4. Summary of the best-generalized linear model (GLM)  (according to AIC) selected for bacterial production. Columns show the estimates, standard errors, Wald statistics, and p-values for the selected predictor variables. The complete set of models with δAIC< 2.0 are shown in Table S4.


| Dependent Variables | Predictor Variables | Estimate | Standard Error | Wald Stat. | p-values |
|---|---|---|---|---|---|
| Bacterial heterotrophic production | Intercept | 5.249544 | 0.334563 | 246.2003 | 0.000000 |
| | Site | --- | --- | 36.5871 | 0.000001 |
| | Salinity | -0.436568 | 0.120411 | 13.1453 | 0.000288 |
| | VA | -0.187402 | 0.066857 | 7.8570 | 0.005062 |