# Peer review of "Alternation of heterotrophic bacterial and archaeal production along nitrogen and salinity gradients in coastal wetlands"

_Biogeosciences, 2020_

## Referee Comment (RC1) · Anonymous Referee #1 · 6 Jun 2020

This paper presents the bacterial and archaeal abundance and heterotrophic production in nine coastal wetlands. Based on Generalized linear models they conclude to switch from heterotrophic bacterial production towards heterotrophic archaeal production as salinity and virus abundance increased. This topic is very interesting in a context of global change. But in my opinion the conclusions are very speculative and based only on linear models between productiity and salinity or viral load. I am not a modeler, but the use of GLP must be justified and statistics must be provided. The experimental methods used are also to be discussed.

Material and Methods Part In my opinion different methods are not very suitable: 1>

Line 145: We obtained the heterotrophic prokaryotic abundance (HPA) by subtracting the cyanobacteria abundance (CyA) from prokaryotic abundance (PA). Cyanobacteria are not the only autotrophic organisms, there are also, for example, nitrifiers. What is the percentage of cyanobacteria? what is the objective in this paper to limit itself to present the HPA numbers? what is the experimental error in the quantification in comparison with the numbers of cyanobacteria? The authors probably wanted to link the number of heterotrophic organisms and their productivity. this is always tricky because the count concerns the total number of organisms without information on the active fraction. 2> line 146: Virus abundance With this protocol and depending on the cytometer used (not specified, but I imagine it is a Beckton Dickinson (BD)), the authors will only see particles >50nm in size and DNA vrus. So they should not say in the text that they will have the actual abundance (since there are many smaller DNA viruses and also RNA viruses).

3> The use of erythromycin to discriminate bacterial versus archaeal production should be discussed. Erythromycin inhibits the growth of bacteria by interfering with protein biosynthesis. It binds with the 50S ribosomal subunit and thus prevents the translocation of peptides and the formation of polypeptides.

efficiency of EMY are related to medically relevant organisms (e.g. Staphylococcus aureus) and do not consider natural prokaryotic assemblages. It is important to note, however, that all other studies concerning the efficiency of EMY are related to medically relevant organisms (e.g. Staphylococcus aureus) and do not consider natural prokaryotic assemblages. Horizontal gene transfer and/or mutations of ribosomal binding sites might alter the sus-ceptibility to EMY in archaeal and bacterial species For Frank 2016, The addition of EMY reduced the bulk leucine incorporation by ∼77%. Evaluation of the inhibition efficiency of EMY on a cell-specific level showed no difference between Archaea (76.0 ± 14.2% [SD]) and Bacteria (78.2 ± 9.5%). Their results suggest that in complex open-ocean prokaryotic communities EMY is efficient as a domain-specific inhibitor Line 160: "it appears to have better efficiencies (ca. 80%) in water of higher

salinity and for specific functional groups as nitrifiers, particularly Firmicutes" I don't understand this sentence that needs to be rephrased, nitrifiers are not included in the firmicutes phylum. It is also necessary to qualify this statement because the authors also mention adaptation and resistance to EMY.

Results part

The results are presented in their entirety by integrating the entire dataset obtained from all 12 sites. For each site there is a strong salinity gradient and also a great heterogeneity in the bacterial numeration and production. Before integrating the whole dataset into a GLM model, the data could be presented and analyzed by station et compared.

Line 215 : significantly , can you give a p-value?

Discussion

Do you have a hypothesis to explain from a physiological point of view the effect of TDN on the switch from heterotrophic bacterial to archaeaous production?

Then the discussion turns to nitrification by archaea, I don't understand the connection since nitrifiers are autotrophic organisms

Line 270 : In our study, ammonia oxidation by archaea during nitrification likely is not a significant process due to the high concentrations of dissolved nitrogen in most wetlands: I don't understand this part of the discussion then nitrifiers are aerobic and except in atypical pathways they need oxygen to achieve nitrification.

More generally, there are only TDN data (including the concentrations of the different organic and inorganic nitrogen forms as well as nitrate, ammonium) and the discussion focuses on the transformation processes between the different oxidation states such as nitrification and denitrification. This seems to me very speculative

the authors state in the final lines of the conclusion: Archaea appeared to be the main

prokaryotes processing nitrogen in the most saline wetlands, I think this is based on a positive correlation between TDN and heterotrophic production by the archaea

Is there a cross-effect between DDT and salinity?

---

## Referee Comment (RC2) · Anonymous Referee #2 · 10 Sep 2020

This manuscript presents an analysis of the salinization effect on the microbial communities' composition and activities. In this study, microbial communities from 112 ponds across the western Mediterranean coast were analyzed based on 13 biotic and abiotic parameters. The salinization effect of the coastal wetland is an important outcome of the sea-level rise and has been directly linked to the ongoing global climate change. Thus, a better understanding of the microbial community response to sea-level rise is an essential step forward in the development of holistic eco-economic models of climate change consequences. The authors concluded that the concentration of Total Dissolved Nitrogen (TDN) positively correlated with the abundances of heterotrophic prokaryotes, but negatively affected the heterotrophic bacterial activity. Additionally, the

authors suggested that a decline in the heterotrophic bacterial activity is due to elevated salinity and higher viral titer. Although these findings are interesting and important to decode the ecosystem response to environmental perturbation, a few methodological and statistical justifications might strengthen the manuscript.

One of the primary authors' conclusion is that heterotrophic bacterial activity is negatively affected by virus titer and salinity. Nevertheless, based on the info in Table 1, in 36.1% (39/108) of the samples, the authors failed to detect virus abundances. On the other hand, the salinity of these samples spends 4 orders of magnitude ranging from 0.2 to 238.8 ppt. Thus, the authors might want to address this disagreement between the major conclusion and the presented data.

The authors used GLM to determine the main drivers of the microbial patterns. One of the primary advantages of the GLM does not need to transform the data to meet the linear model assumptions. Instead, GLM analysis allows modifying the model assumptions, thus that it is not clear why the authors applied data transformations (line 177-180). Additionally, the model selection based ACI may be problematic or even inaccurate when compare the models of transformed/modified data and original datasets. Finally, to increase the readability and reproductivity of the data analysis, the author might include the chosen model assumptions in the method section.

To quantify the fraction of different microbial classes, the author used fluorescence labeling follows by FACS counting. This is a powerful technique when applied to fresh samples. In this study, the authors feezed the sample in liquid nitrogen and stored at -80C until analysis (lines 134-137). Frequently, freezing the bacteria cell leads to cell disruption and DNA release, unless the protective reagents such glycerol were introduced prior to freezing. The molecular probe, Cyber Green I, which was used in this study, frequently fails to distinguish between environmental and cellular DNA. Moreover, Cyber Green I, equally labels eukaryotic, prokaryotic, and environmental cells, which may introduce biases into data interpretation. I aware that it might be impossible to repeat the cells counting; nevertheless, the authors should insert the

appropriate correction to the manuscript.

The authors calculated the abundances of the heterotrophic prokaryotes by subtracting cyanobacterial abundance from prokaryotic abundance (lines 145-146). The authors might extend the discussion about this approach since cyanobacteria are not the only ones with autotrophic capacities in the system, other non-photosynthetic autotrophs are involved in sulfur, iron, and nitrogen transformation might play an essential role in the coastal ecosystem. Moreover, many cyanobacterial strains exhibit a multicellular lifestyle, growing as filaments that can be hundreds of cells long and endowed with intercellular communication. Thus, it is crucial to clarify how exactly cyanobacteria were counted.

Throughout the manuscript, the authors use the term "production"; I find this term misleading. The biological production usually refers to primary productivity; in this study, the authors applied leucine incorporation assay to measure protein synthesis or community activity. To increase the manuscript readability, the authors might want to replace the "production" to "activity" as it was written inline 277.

To distinguish between bacterial and archaeal activities, the authors applied erythromycin, which binds to the 23S rRNA component of the 50S ribosome and interferes with the assembly of 50S subunits. Although usage of erythromycin is a common practice to limit bacterial protein synthesis, however many bacteria have natural erythromycin resistance. Moreover, since erythromycin blocks mainly bacterial protein synthesis and has a limited effect on eukaryotic activities, based on the presented data, I not sure for what extend the signal recorded in this study is a result of bacterial, archaeal, or eukaryotic protein synthesis. Thus, the authors might want to clarify the methodological limitation of erythromycin usage in this study.

Please provide the statistical support, including p-values and $R^2$, for the data presented in figures 4, 5, and 6.

---

## Author Comment (AC1) · 30 Sep 2020

Please see below the responses to the Anonymous Referee #1 and the actions taken regarding her/his concerns.

In the text below, the suggestions and comments of the Anonymous Referee #1 are in black and plain font and *our responses are in italics and blue font*.

Anonymous Referee #1

This paper presents the bacterial and archaeal abundance and heterotrophic production in nine coastal wetlands. Based on Generalized linear models they conclude to switch from heterotrophic bacterial production towards heterotrophic archaeal production as salinity and virus abundance increased. This topic is very interesting in a context of global change. But in my opinion the conclusions are very speculative and based only on linear models between productivity and salinity or viral load. I am not a modeler, but the use of GLP must be justified and statistics must be provided. The experimental methods used are also to be discussed.

Material and Methods

Part In my opinion different methods are not very suitable:

1> Line 145: We obtained the heterotrophic prokaryotic abundance (HPA) by subtracting the cyanobacteria abundance (CyA) from prokaryotic abundance (PA). Cyanobacteria are not the only autotrophic organisms, there are also, for example, nitrifiers. What is the percentage of cyanobacteria? what is the objective in this paper to limit itself to present the HPA numbers? what is the experimental error in the quantification in comparison with the numbers of cyanobacteria? The authors probably wanted to link the number of heterotrophic organisms and their productivity. This is always tricky because the count concerns the total number of organisms without information on the active fraction.

*We agree with the reviewer and thank this comment. In the new version, we have only considered the total prokaryotic abundance and the free, non-colonial cyanobacteria abundance. As the reviewer noted, nitrifiers are included in the autotrophic fraction. We have changed all the analysis that included the "heterotrophic prokaryotic abundance (HPA)" for the "prokaryotic abundance (PA)".*

*Cyanobacteria cells were counted using a different sample and with different cytometer conditions (please see the method section). They only represent free-living cells; colonial filaments or aggregates are not included.*

*As the reviewer detected, we initially wanted to relate "heterotrophic prokaryotic abundance" with heterotrophic activity. Therefore, we subtracted the cyanobacteria cells from the total pool; however, we did not consider other autotrophic prokaryotes such as nitrifiers, and this calculation was not accurate. Now, we have presented only the total abundance of prokaryotes.*

*We have deleted all the statistical analyses, figures, and tables that included the "heterotrophic prokaryotic abundance (HPA)". Now, we have included similar analyses but using the total abundance of prokaryotes. The main message of the MS is still the same.*

2> line 146: Virus abundance With this protocol and depending on the cytometer used (not specified, but I imagine it is a Beckton Dickinson (BD)), the authors will only see particles >50nm in size and DNA virus. So they should not say in the text that they will have the actual abundance (since there are many smaller DNA viruses and also RNA viruses).

*Indeed, the flow cytometer was a Beckton Dickinson FACSCalibur (Franklin Lakes, NJ, USA). Now, we have included this information in the methods.*

*We have now explicitly mentioned that these virus abundances represent minimum estimates of viral abundance in the methods section. Currently, there are no accepted approaches for direct counts of viruses containing RNA or double-stranded DNA for natural waters.*

3> The use of erythromycin to discriminate bacterial versus archaeal production should be discussed. Erythromycin inhibits the growth of bacteria by interfering with protein biosynthesis. It binds with the 50S ribosomal subunit and thus prevents the translocation of peptides and the formation of polypeptides. The efficiency of EMY are related to medically relevant organisms (e.g. Staphylococcus aureus) and do not consider natural prokaryotic assemblages. It is important to note, however, that all other studies concerning the efficiency of EMY are related to medically relevant organisms (e.g. Staphylococcus aureus) and do not consider natural prokaryotic assemblages. Horizontal gene transfer and/or mutations of ribosomal binding sites might alter the susceptibility to EMY in archaeal and bacterial species For Frank 2016,The addition of EMY reduced the bulk leucine incorporation by 77%. Evaluation of the inhibition efficiency of EMY on a cell-specific level showed no difference between Archaea (76.0 ± 14.2% [SD]) and Bacteria (78.2 ± 9.5%). Their results suggest that in complex open-ocean prokaryotic communities EMY is efficient as a domain-specific inhibitor Line 160: "it appears to have better efficiencies (ca. 80%) in water of higher salinity and for specific functional groups as nitrifiers, particularly Firmicutes" I don't understand this sentence that needs to be rephrased, nitrifiers are not included in the firmicutes phylum. It is also necessary to qualify this statement because the authors also mention adaptation and resistance to EMY.

*We agree with the reviewer. This sentence was misleading, and we thank this comment. The study of Du et al. (2016) showed that nitrifiers (i.e., ammonia-oxidizing bacteria (AOB) and nitrite-oxidizing bacteria (NOB)) were susceptible to erythromycin, particularly the NOB. Moreover, other gram-positive bacteria that belong to the Firmicutes Phylum are also particularly sensitive. We have now rewritten this sentence to avoid this misunderstanding.*

Results part
The results are presented in their entirety by integrating the entire dataset obtained from all 12 sites. For each site there is a strong salinity gradient and also a great heterogeneity in the bacterial numeration and production. Before integrating the whole dataset into a GLM model, the data could be presented and analyzed by station et compared.

*In the new version, we have included simple regressions for each site in the supplementary information. The patterns were significant and consistent for the sites that included high numbers of ponds. We included this information about the site-specific relationships in the text (please see new results). However, we consider that the figures should appear only as supplementary information to make the paper more readable.*

Line 215 : significantly , can you give a p-value?

*We have now included in the text the p-value and $R^2$.*

Discussion
Do you have a hypothesis to explain from a physiological point of view the effect of TDN on the switch from heterotrophic bacterial to archaeaous production?
Then the discussion turns to nitrification by archaea, I don't understand the connection since nitrifiers are autotrophic organisms.-
Line 270 : In our study, ammonia oxidation by archaea during nitrification likely is not a significant process due to the high concentrations of dissolved nitrogen in most wetlands: I don't understand this part of the discussion then nitrifiers are aerobic and except in atypical pathways they need oxygen to achieve nitrification.

*We agree with the reviewer, and we have deleted this paragraph in the new version of the manuscript. We have mostly focused the discussion about the denitrification by archaea and heterotrophic activity across the salinity gradient.*

More generally, there are only TDN data (including the concentrations of the different organic and inorganic nitrogen forms as well as nitrate, ammonium) and the discussion focuses on the transformation processes between the different oxidation states such as nitrification and denitrification. This seems to me very speculative the authors state in the final lines of the conclusion: Archaea appeared to be the main prokaryotes processing nitrogen in the most saline wetlands, I think this is based on a positive correlation between TDN and heterotrophic production by the archaea

*We agree with the reviewer and have tone down the conclusions. We have deleted the paragraph about ammonia-oxidizing archaea since we only have data on heterotrophic archaeal activity.*

Is there a cross-effect between DDT and salinity?

*Assuming the reviewer means TDN instead of DDT. There is a correlation between TDN and salinity (r= 0.33, p <0.001). This correlation is one of the reasons for using GLMs. The best model included TDN, instead salinity, as the main driver for the case of archaeal activity.*

---

## Author Comment (AC2) · 30 Sep 2020

Please see below the responses to the Anonymous Referee #2 and the actions taken regarding her/his concerns.
In the text below, the suggestions and comments of the Anonymous Referee #2 are in black and plain font and *our responses are in italics and blue font*.

This manuscript presents an analysis of the salinization effect on the microbial communities' composition and activities. In this study, microbial communities from 112 ponds across the western Mediterranean coast were analyzed based on 13 biotic and abiotic parameters. The salinization effect of the coastal wetland is an important outcome of the sea-level rise and has been directly linked to the ongoing global climate change. Thus, a better understanding of the microbial community response to sea-level rise is an essential step forward in the development of holistic eco-economic models of climate change consequences. The authors concluded that the concentration of Total Dissolved Nitrogen (TDN) positively correlated with the abundances of heterotrophic prokaryotes, but negatively affected the heterotrophic bacterial activity. Additionally, the authors suggested that a decline in the heterotrophic bacterial activity is due to elevated salinity and higher viral titer. Although these findings are interesting and important to decode the ecosystem response to environmental perturbation, a few methodological and statistical justifications might strengthen the manuscript.
One of the primary authors' conclusion is that heterotrophic bacterial activity is negatively affected by virus titer and salinity. Nevertheless, based on the info in Table 1, in 36.1% (39/108) of the samples, the authors failed to detect virus abundances. On the other hand, the salinity of these samples spends 4 orders of magnitude ranging from 0.2 to 238.8 ppt. Thus, the authors might want to address this disagreement between the major conclusion and the presented data.

*We are sorry about this misunderstanding. In Table 1, "n. d." means "not determined" but not "not detected". Unfortunately, we could not count the samples of viruses from the Camargue, Sardinia, and Tunisia wetlands due to preservation problems during the travels. Now, we have included the meaning of "n.d." in the legend of Table 1 to avoid this misunderstanding among the readers. That is, there is not a disagreement between the conclusions and the data. It was a pitfall that we did not include the meaning of "n.d." in the previous version of this MS.*

The authors used GLM to determine the main drivers of the microbial patterns. One of the primary advantages of the GLM does not need to transform the data to meet the linear model assumptions. Instead, GLM analysis allows modifying the model assumptions, thus that it is not clear why the authors applied data transformations (line 177-180). Additionally, the model selection based ACI may be problematic or even inaccurate when compare the models of transformed/modified data and original datasets.
Finally, to increase the readability and reproductivity of the data analysis, the author might include the chosen model assumptions in the method section.

*We used an identity link and a normal error distribution, which we will now specify in the text. We respectfully disagree with the other statements. GLMs are more flexible than traditional regression methods, but they still will produce spurious results unless the precise nature of the data, the link function, and the error function are selected with care. Just by conducting a GLM there is no guarantee that the models will reliable. In our case, the nature of the data mean that use of a normal error distribution and inspection of model residuals to identify the*

*transformation that removes heteroscedasticity is a valid approach. Using these kind of GLMs without transformations produces strong heteroscedasticity, so the results are entirely unreliable with a very strong influence of samples that are outliers. Using AIC for model selection does not solve that problem, and similarly the P values would not be meaningful. We selected those transformations that were required to remove heteroscedasticity, as already explained in the methods of the manuscript.*

To quantify the fraction of different microbial classes, the author used fluorescence labeling follows by FACS counting. This is a powerful technique when applied to fresh samples. In this study, the authors feezed the sample in liquid nitrogen and stored at -80C until analysis (lines 134-137). Frequently, freezing the bacteria cell leads to cell disruption and DNA release, unless the protective reagents such glycerol were introduced prior to freezing. The molecular probe, Cyber Green I, which was used in this study, frequently fails to distinguish between environmental and cellular DNA. Moreover, Cyber Green I, equally labels eukaryotic, prokaryotic, and environmental cells, which may introduce biases into data interpretation. I aware that it might be impossible to repeat the cells counting; nevertheless, the authors should insert the appropriate correction to the manuscript.

*We have used well-established flow cytometry procedures for counting prokaryotes and viruses (i.e., Gasol & del Giorgio 2000; Brussard et al. 2010), trying to be extremely careful. We are confident about the performance of these analyses and the data obtained. Usually, the side scatter (SSC) is different for each size group. Viruses, prokaryotes, and eukaryotes have different SSC windows. We selected different SSC windows at the beginning of the counting process for viruses and prokaryotes that we kept them all the time. Free-living, non-colonial cyanobacteria can be discriminated of the prokaryotes pool by their content in chlorophyll a (red fluorescence) in the FL3. Please see the figure below. Usually, environmental DNA is out of these SSC windows, in the noisy region. We have now included these details in the method section.*

[Figure]

The authors calculated the abundances of the heterotrophic prokaryotes by subtracting cyanobacterial abundance from prokaryotic abundance (lines 145-146). The authors might extend the discussion about this approach since cyanobacteria are not the only ones with autotrophic capacities in the system, other non-photosynthetic autotrophs are involved in sulfur, iron, and nitrogen transformation might play an essential role in the coastal ecosystem. Moreover, many cyanobacterial strains exhibit a multicellular lifestyle, growing as

filaments that can be hundreds of cells long and endowed with intercellular communication. Thus, it is crucial to clarify how exactly cyanobacteria were counted.

*We agree with the reviewer that there are more autotrophic prokaryotes besides cyanobacteria, such as nitrifiers and other groups that we did not consider in the previous version. Therefore, the calculation of the "heterotrophic prokaryotic abundance (HPA)" was not correct. In the new version, we have included only the total abundance of prokaryotes. Please see also the reply to reviewer #1 on this point.*
*Cyanobacteria cells were counted using a different sample and with different cytometer conditions (please see the method section). They only represent free-living cells; colonial filaments or aggregates were not included.*
*We have deleted all the statistical analyses, figures and tables that included the "heterotrophic prokaryotic abundance (HPA)". Now, we have included similar analyses but using the total abundance of prokaryotes (i.e., without cyanobacteria subtraction). The main message of the MS is still the same.*

Throughout the manuscript, the authors use the term "production"; I find this term misleading. The biological production usually refers to primary productivity; in this study, the authors applied leucine incorporation assay to measure protein synthesis or community activity. To increase the manuscript readability, the authors might want to replace the "production" to "activity" as it was written inline 277.

*Bacterial production and bacterial activity (protein synthesis) can be both found in the scientific literature (see, for instance, Cole et al. 1988; Kirchman et al. 1985). However, we have followed the reviewer suggestion and changed "production" by "activity" to make the MS more readable.*

To distinguish between bacterial and archaeal activities, the authors applied erythromycin, which binds to the 23S rRNA component of the 50S ribosome and interferes with the assembly of 50S subunits. Although usage of erythromycin is a common practice to limit bacterial protein synthesis, however many bacteria have natural erythromycin resistance. Moreover, since erythromycin blocks mainly bacterial protein synthesis and has a limited effect on eukaryotic activities, based on the presented data, I not sure for what extend the signal recorded in this study is a result of bacterial, archaeal, or eukaryotic protein synthesis. Thus, the authors might want to clarify the methodological limitation of erythromycin usage in this study.

*We have included more details about erythromycin susceptibility of specific bacteria functional groups related to the nitrogen cycle (i.e., ammonia-oxidizing bacteria (AOB) and nitrite-oxidizing bacteria (NOB)) or other Phyla such as Firmicutes.*

Please provide the statistical support, including p-values and $R^2$, for the data presented in figures 4, 5, and 6.

*We have now included the statistical support (p-value and R2) for the data presented in the figures 4, 5 and 6.*